# Analysis of Extracellular Vesicles in Gastric Juice from Gastric Cancer Patients

**DOI:** 10.3390/ijms20040953

**Published:** 2019-02-22

**Authors:** Shuji Kagota, Kohei Taniguchi, Sang-Woong Lee, Yuko Ito, Yuki Kuranaga, Yasuyuki Hashiguchi, Yosuke Inomata, Yoshiro Imai, Ryo Tanaka, Keitaro Tashiro, Masaru Kawai, Yukihiro Akao, Kazuhisa Uchiyama

**Affiliations:** 1Department of General and Gastroenterological Surgery, Osaka Medical College, 2-7 Daigaku-machi, Takatsuki, Osaka 569-8686, Japan; sur165@osaka-med.ac.jp (S.K.); sur079@osaka-med.ac.jp (S.-W.L.); sur171@osaka-med.ac.jp (Y.I.); sur141@osaka-med.ac.jp (Y.I.); sur148@osaka-med.ac.jp (R.T.); sur085@osaka-med.ac.jp (K.T.); sur080@osaka-med.ac.jp (M.K.); uchi@osaka-med.ac.jp (K.U.); 2Translational Research Program, Osaka Medical College, 2-7 Daigaku-machi, Takatsuki, Osaka 569-8686, Japan; 3Department of Anatomy and Cell Biology, Division of Life Sciences, Osaka Medical College, 2-7 Daigaku-machi, Takatsuki, Osaka 569-8686, Japan; an1006@art.osaka-med.ac.jp; 4United Graduate School of Drug Discovery and Medical Information Sciences, Gifu University, 1-1 Yanagido, Gifu 501-1193, Japan; v3501001@edu.gifu-u.ac.jp (Y.K.); yakao@gifu-u.ac.jp (Y.A.); 5Department of Biology, Osaka Medical College, 2-7 Daigaku-machi, Takatsuki, Osaka 569-8686, Japan; bio007@osaka-med.ac.jp

**Keywords:** exosome, microvesicles, extracellular vesicles, gastric juice, gastric cancer

## Abstract

Extracellular vesicles (EVs) are secretory membrane vesicles containing lipids, proteins, and nucleic acids; they function in intercellular transport by delivering their components to recipient cells. EVs are observed in various body fluids, i.e., blood, saliva, urine, amniotic fluid, and ascites. EVs secreted from cancer cells play important roles in the formation of their environment, including fibrosis, angiogenesis, evasion of immune surveillance, and even metastasis. However, EVs in gastric juice (GJ-EVs) have been largely unexplored. In this study, we sought to clarify the existence of GJ-EVs derived from gastric cancer patients. GJ-EVs were isolated by the ultracentrifuge method combined with our own preprocessing from gastric cancer (GC) patients. We verified GJ-EVs by morphological experiments, i.e., nanoparticle tracking system analysis and electron microscopy. In addition, protein and microRNA markers of EVs were examined by Western blotting analysis, Bioanalyzer, or quantitative reverse transcription polymerase chain reaction. GJ-EVs were found to promote the proliferation of normal fibroblast cells. Our findings suggest that isolates from the GJ of GC patients contain EVs and imply that GJ-EVs partially affect their microenvironments and that analysis using GJ-EVs from GC patients will help to clarify the pathophysiology of GC.

## 1. Introduction

Intercellular communication is an essential hallmark of multicellular organisms and can be mediated through direct cell–cell contact or the transferal of secreted molecules. Extracellular vesicles (EVs) play important roles in cell–cell communication by shuttling various molecules from donor to recipient cells [1,2]. In the early 1980s, EVs were found and initially thought to be a means for cells to expel membranous debris [3]. Over the last three decades, a mechanism for intercellular communication involving intercellular transferal of EVs has become evident [4,5,6]. EVs are membranous vesicular structures existing as different vesicle types, including shedding microvesicles (MVs), exosomes, and apoptotic bodies. MVs are (100–1000 nm) membranous vesicles derived from outward blebbing of cell surfaces. Exosomes are smaller vesicles (40–100 nm) released by exocytosis of multivesicular bodies (MVBs) from a variety of cells [1,7]. EVs have been isolated from diverse body fluids such as blood [8], urine [9], breast milk [10], saliva [11], and cerebrospinal fluid [12]. Also, EVs contain various functional molecules such as microRNAs, mRNAs, DNAs, lipids, and proteins [7]. EVs act as intercellular transporters by delivering their gene components to recipient cells. In particular, cancer-cell-derived EVs play a crucial role in the formation of their environment by involvement in activities such as fibrosis, angiogenesis, evasion of immune surveillance, and metastasis [13,14,15]. Therefore, genetic information of EVs is expected to serve as new biomarkers of various cancers. However, EVs contained in gastric juice (GJ-EVs) have been largely unstudied. 

In fact, the management, including the diagnosis and treatment of gastric cancer (GC) patients, remains a clinical problem, and several reports have suggested that exosomes in the blood have the potential of becoming new and early diagnostic biomarkers even in GC [16,17]. Hence, further elucidation of GC pathology including EVs is required. In this study, we sought to clarify the existence of GJ-EVs from GC patients in order to develop a better understanding of the pathologic conditions of GC. 

## 2. Results

### 2.1. A Preprocessing Step was Established to Isolate Extracellular Vesicles (EVs) from Gastric Juice (GJ)

Firstly, we tried to isolate EVs from the GJ of GC patients by using the ultracentrifuge method. However, transmission electron microscopy (TEM) showed that these EVs were covered with abundant mucin (Figure 1A). Interestingly, some of these EVs were positive for CD63, which is known as a representative exosomal marker [7] (Figure 1B). Hence, we established a preprocessing method for use prior to ultracentrifugation in order to isolate EVs from GJ (Figure 2). 

### 2.2. Isolates from GJ of Gastric Cancer (GC) Patients Contained EVs

We examined the size-population in our isolated materials from GJ by performing nanoparticle tracking analysis (NTA). As a result, NTA showed that the peak of the isolated EVs was around 140–200 nm (Figure 3A, Appendix A, and Table 1). Also, these EVs, which were bound to microbeads, were clearly observed by scanning electron microscope (SEM) analysis (Figure 3B). SEM micrographs showed EVs of various sizes less than 1000 nm (Figure 3C). These findings implied that our purification method could collect homogenous EVs from GJ.

### 2.3. Protein Markers of Extracellular Vesicles (EVs) were Detected in GJ-EVs

Next, we investigated protein expression profiles of our isolated EVs by performing Western blotting analysis. TSG101, CD9, and CD81 are well known as EV markers [7], and GOLGA2 (GM130) and CANX (calnexin) as negative markers of EVs [18]. Firstly, we examined the expression profiles of these protein markers in EVs isolated from GC cell lines (KATOIII and MKN7). As shown in Figure 4A, all positive markers of EVs were detected in isolates from these GC cell lines. As expected, the negative markers were not detected (Figure 4A). Also, the same tendency was observed in isolates from GJ, although some differences existed among the samples examined (Figure 4B). These findings suggested that our isolates from GJ contained EVs.

### 2.4. EVs from GJ Contained Several MicroRNAs

Moreover, we investigated the existence of microRNAs (miRNAs), which are reportedly present in EVs [7,19], in our isolates from GJ. In order to examine the expression of miRNAs, we isolated RNA from samples by using a RNA 6000 Pico Kit. Results from the Bioanalyzer showed that our EVs contained RNAs comprising 25 to 200 nucleotides and showed no signs of ribosomal 18S or 28S peaks (Figure 5, Appendix A, and Table 2). Also, the peak of RNA was around 25 nucleotides in all GC cases tested (Figure 5, Appendix A, and Table 2). Next, we examined the expression levels of several miRNAs and small nucleolar RNA (snoRNA) in EVs from GJ of 4 GC patients by using RT-qPCR. Based on several reports, MIRLET7A1-5p, MIR16-5p, MIR103a-3p, MIR191-5p, MIR423-5p, and *RNU6-6P*, which are expressed in EVs from serum and cerebrospinal fluid, were selected [20,21,22,23]. As a result, these microRNAs, especially MIR16-5p and MIR191-5p, were present in EVs from the GJ of GC patients (Table 3). These findings suggested that our isolates from GJ were EVs and that miRNAs were one type of genes in them. 

### 2.5. GJ-EVs from GC Patients Promoted Tissue Fibrosis

Finally, in order to examine the functions of GJ-EVs, we exposed ASF-4 cells (normal fibroblast) to GJ-EVs from GC patients and to EVs from KATOIII cells. Our results showed that GJ-EVs were incorporated into ASF-4 cells (Figure 6A and Appendix A). Also, ASF-4 cell growth was promoted by both (KATOIII and GJ-EVs) isolates tested (Figure 6B). These findings suggested that GC cells partially formed a conductive microenvironment through the functions of GJ-EVs (Figure 7).

## 3. Discussion

Recently, EVs have been expected to be useful as a new biomarker [24]. Despite an increasing number of reports related to EVs in body fluids such as blood serum and urine, analysis of GJ-EVs has not been adequate, with only a few reports about them [25]. The main reason is that the isolation of GJ-EVs is relatively difficult because these vesicles are covered with mucus. In this study, we considered that the ultracentrifuge method would be more suitable for collecting GJ-EVs than other methods, such as affinity-based [26] and size-exclusion chromatography [27] ones, because of their viscosity. However, only normal ultracentrifugation was not adequate to isolate pure EVs from GJ. Thus, high-speed/long hours’ ultracentrifugation combined with original preparation of GJ was performed. Importantly, our experiments were advanced after we confirmed the adaptation of our conditions of ultracentrifugation for isolating EVs from cancer cell lines. Further optimization is required, but our methods used in this study could separate EVs from mucus and remove the latter before ultracentrifugation (Figure 2). Mucus is produced by mucous neck cells and covers the mucosa of the stomach in order to protect it from acid, pepsin, and mechanical damage [28]. Therefore, EVs may require a protection system, such as mucus, in order to maintain their functions in such a severe environment. 

In this study, we verified that our isolates from GJ comprised EVs based on the results of various objective experiments. Firstly, NTA indicated that our isolates from GJ constituted relatively homogeneous granules (around 140–200 nm in diameter, Figure 3A and Table 1), and SEM showed that they could be roughly classified into two sizes, i.e., exosome and MV (Figure 3B,C). Also, Western blot analysis showed that tetraspanins (surface markers of EVs) and other markers of EVs were expressed on our isolates. The lack of expression of GOLGA2 and CANX, which are known as negative markers of EVs, strongly supported the existence of GJ-EVs in our isolates (Figure 4). Moreover, Bioanalyzer and RT-qPCR indicated that they contained miRNAs, which are one type of representative gene of EVs (Figure 5, Table 2, and Table 3). Based on these findings, we concluded that our isolates from GJ contained EVs. Interestingly, our results indicated that MIR16-5p and MIR191-5p were expressed well constantly (Table 3). Therefore, these miRNAs are possible real reference genes of EVs. On the other hand, the RNA concentration and the expression levels of tested miRNA in case fifteen were extremely high compared with those in the other cases (Table 2 and Table 3). These results suggested the relevancy of cancer-cell-derived EVs as clinical and pathological characteristics (Table 4). Namely, abundant EVs might be released in the advanced stage of type 4 GC, especially in the case of signet-ring cell carcinoma. Of course, our study had several limitations, such as the number of patients, and further investigation is needed in order to establish GJ-EVs as a biomarker of GC. The most essential experiment is to compare the genes contained in GJ-EVs between GC patients and healthy donors. 

GC is classified into six types. In particular, type four GC, which begins in the lining of the stomach and spreads to the stomach wall, has an extremely poor prognosis. Detecting type four GC in its early stage is very difficult by endoscopy. This is one of the significant issues of GC. Also, fibroblasts surrounding cancer cells, known as cancer-associated fibroblasts (CAFs), play important roles in the progression and growth of cancer cells, including GC [29,30]. Especially in type four GC, cancer-stroma cells protect against the permeation of anticancer drugs [31]. In addition, the proliferative and invasive abilities of type 4 GC cells are closely associated with growth factors produced by CAFs [32,33]. The results of our cell viability experiment, using normal fibroblasts cells, suggested that GJ-EVs may function as a coordinator of their microenvironment, forming conditions preferable to cancer cells (Figure 6). A further detailed investigation is required to determine the mechanisms by which GJ-EVs affect the cancer microenvironment in GC patients. For example, more detail regarding the characteristics of fibroblasts as CAFs should be investigated. We expect that GJ-EVs will be one of the key materials in order to elucidate GC conditions and also that GJ-EVs will be established as a new biomarker of GC, especially for type four GC.

## 4. Materials and Methods

### 4.1. Patients and Samples

All human gastric juice samples were obtained from patients who had undergone surgery at Osaka Medical College Hospital (Takatsuki, Osaka, Japan). Before their operation, patients were managed without food for approximately half a day and without water for 2 h. Collection and investigation of the samples from the patients were approved by the research ethics committee of Osaka Medical College (approval number: 1641, 1 January 2015). Twelve patients with previously untreated GC and 6 patients with neoadjuvant chemotherapy were selected. Detailed clinical information is shown in Table 4.

### 4.2. Cell Lines and Cell Culture

Human GC cell lines KATO-III and MKN-7 were gifts from Nobuhiko Tanigawa (an Emeritus Professor of our department), and human normal fibroblasts cells ASF-4 were obtained from the JCRB (Japanese Collection of Research Bio Resources) cell bank. GC cells were cultured under a 95% air/5% CO_2_ atmosphere at 37 °C in RPMI-1640 (Invitrogen, Carlsbad, CA, USA) supplemented with 10% Exosome-depleted FBS (System Biosciences, LLC, Palo Alto, CA, USA). In the case of ASF-4 cells, MEM (Sigma-Aldrich Co. LLC, St. Louis, MO, USA) was used as the culture medium. 

### 4.3. Isolation of Cancer-Cell-Line-Derived EVs

For the collection of GC-cell-line-derived EVs, GC cells were cultured in large flasks. After that, the culture medium was collected and centrifuged at 2000 rpm for 5 min. The resulting supernatant was filtered through a 0.45-mm pore filter to remove cellular debris and then through a 0.22-mm pore filter to isolate EVs and/or apoptotic bodies. The flow-through fraction was ultracentrifuged at 100,000 rpm (450,200× *g*) for 3 h [34,35]. Without disturbing the EV pellet, the supernatant was carefully removed. Then, the EVs were collected from the resulting pellet, washed with phosphate-buffered saline (PBS), and stored at −80 °C.

### 4.4. Sample Preparation for Isolation of EVs

Before ultracentrifugation, our own preprocessing step was performed for GJ. GJ was filtered through a 0.23-mm mesh for removing residues. After that, mucus on the mesh was grinded with a surgical knife in order to divide it into pieces. The shredded mucus and previous filtrate were collected and passed through a 0.22-μm pore filter. The filtrate was then ultracentrifuged at 100,000 rpm (450,200× *g*) for 3 h. Without disturbing the pellet, the supernatant was carefully removed, and the nanoparticles were then collected from the resulting pellet, washed with PBS, and stored at −80 °C (Figure 2).

### 4.5. Electron Microscopic Analysis

Isolates from GJ were harvested and rinsed with PBS. They were fixed for 2 h with 2% paraformaldehyde and 2.5% glutaraldehyde in 0.1 M phosphate buffer (pH 7.4, PB), rinsed in PB, and subsequently fixed in 1% osmium OsO_4_ for 2 h. After having been washed with PB, the samples were dehydrated in a series of graded ethanol concentrations and then cleared in propylene oxide and embedded in epoxy resin mixture. Thereafter, ultrathin sections (70-nm thickness) were prepared, after which they were stained with uranyl acetate and lead citrate and examined by transmission electron microscope (TEM) with a Hitachi-7650 (Hitachi, Tokyo, Japan). In the case of the scanning electron microscopic (SEM) analysis, isolates from GJ were attached to 3-µm size microbeads. After the same process for TEM, the GJ-microbeads were coated with platinum (Pt)-carbon. Finally, the samples were examined with a Hitachi S-5000 (Hitachi, Tokyo, Japan).

### 4.6. Nano Tracking Analysis

Nanoparticle tracking analysis (NTA) is a method used for detecting secreted nanoparticles in a liquid sample. GJ-EVs suspended in 1 mL of PBS were analyzed by a NTA Version 2.3 Build 0034 instrument (Nanosight, Wiltshire, UK). The following photographic conditions were used: frames processed (1498 of 1498 or 1499 of 1499); frames per second (24.97 or 24.98); calibration (190 nm/pixel); and detection threshold (6 or 7 multi). Samples were diluted at 1:1000 in PBS and analyzed [34,35,36].

### 4.7. Western Blot Analysis

Whole isolates were lysed with chilled radioimmunoprecipitation assay (RIPA) buffer (Thermo Fisher Scientific Inc., Waltham, MA, USA) and 1% protease inhibitor cocktail (Sigma-Aldrich Co. LLC, St. Louis, MO, USA) and stood for 15 min on ice. After centrifugation at 12,000 rpm for 20 min at 4 °C, the supernatants were collected as whole-isolate protein samples. Protein contents were measured with a DC Protein assay kit (BIO-RAD, Laboratories, Inc., Hercules, CA, USA). Ten micrograms of lysate protein was separated by SDS-PAGE using 7.5–12.5% polyacrylamide gels (Wako Pure Chemical Industries, Ltd., Osaka, Japan), and electroblotted onto a PVDF membrane (PerkinElmer Life Sciences, Inc., Waltham, MA, USA). After blockage of nonspecific binding sites for 1 h with 5% nonfat milk (Cell Signaling Technology, Inc., Danvers, MA, USA) in PBS containing 0.05% Tween 20 (PBS-T), the membrane was incubated overnight at 4 °C with primary antibodies. Primary antibodies were diluted in Can Get Signal^®^ Immunoreaction Enhancer Solution (TOYOBO Co., Ltd., Tokyo, Japan). The following primary antibodies were used: anti-TSG101 (Abcam, Cambridge, UK); anti-CD9 and -CD81, (Santa Cruz Biochemistry, Inc., Dallas, TX, USA); and Calnexin, GM130, β-actin (Cell Signaling Technology, Inc., Danvers, MA, USA). Horseradish peroxidase (HRP)-conjugated horse anti-mouse and -rabbit IgG (Cell Signaling Technology, Inc., Danvers, MA, USA) were used as secondary antibodies. The next day, the membrane was then washed with PBS-T, incubated further for 1 h with secondary antibodies, and then washed with PBS-T. The immunoblots were visualized by use of Luminata TM Forte Western HRP Substrate (Millipore Corporation, Billerica, MA, USA) [37]. Detection and quantification of bands were performed by using an LAS-3000 (Fujifilm, Tokyo, Japan). 

### 4.8. Bioanalyzer

RNA quality and concentration were assessed by using the 2100 Bioanalyzer (Agilent Technologies, Inc., Santa Clara, CA, USA). RNA samples of nanoparticles from GJ were prepared by using an RNA 6000 Pico Kit, and samples were prepared according to the manufacturer’s instructions (Agilent Technologies, Inc., Santa Clara, CA, USA). The analysis shows the EVs-RNA yield and size distribution.

### 4.9. Real-Time Quantitative Reverse Transcription PCR

Total RNA was isolated from GJ by using a mirVana miRNA isolation kit (Invitrogen, Carlsbad, CA, USA) according to the manufacturer’s protocol. RNA concentrations and purity were assessed by the Bioanalyzer. We conducted quantitative RT-PCR by using TaqMan^®^ MicroRNA Reverse Transcription Kit (Applied Biosystems, Foster City, CA, USA) and THUNDERBIRD Probe qPCR Mix (TOYOBO Co., Ltd., Osaka, Japan) according to the manufacturer’s protocol. The primers for MIRLET7A-5p, MIR16-5p, MIR103a-3p, MIR191-5p, MIR423-5p, and *RNU6-6P* were purchased from TaqMan^TM^ MicroRNA Assays (Applied Biosystems, Foster City, CA, USA). The relative expression levels were calculated by the ΔΔ*C*_t_ method. 

### 4.10. Cell Viability Experiment

Human normal fibroblastic ASF-4 cells were seeded in 6-well plates at a concentration of 0.5 × 10^5^/mL in each well on the day before exposure to EVs from GJ. The EVs were derived from KATO-III cells and the GJ from GC patients. Penicillin–streptomycin (FUJIFILM Wako Pure Chemical Corporation, Osaka, Japan) was added to all EV samples. Each pellet of EVs was dissolved in PBS and adjusted to the density (5000 and 1250 μg/mL). ASF-4 cells were exposed to 100 µL of PBS containing EVs. Treatment with only PBS containing antibiotics was used as the control. The number of viable cells was determined by performing the trypan-blue dye exclusion test at 72 h after exposure of the cells to the EVs. The values were presented as the mean ± standard deviation. A *p* value < 0.05 was considered to be statistically significant.

### 4.11. Immunofluorescent Microscopy

Mucus from gastric juice was stretched on a silane-coated glass slide (DAKO an Agilent Technologies, Inc. Santa Clara, CA, USA) and then fixed with 4% paraformaldehyde. The mucus was thereafter immunostained with anti-CD63 rabbit polyclonal antibody (Santa Cruz Biochemistry, Inc., Dallas, TX, USA) and Alexa 594 anti-rabbit chicken polyclonal antibody (Life Technologies Corporation, Carlsbad, CA, USA). Samples were observed by laser microscope (Leica TCS SP8, Leica MYCROSYSTEMS, Wetzlar, Germany).

### 4.12. EVs-Incorporation Experiment

ASF-4 cells were seeded in a 4-chamber slide (Nunc™ Lab-Tek™ Chamber Slide System, Thermo Fisher Scientific Inc., Waltham, MA, USA) at a concentration of 0.5 × 10^5^/mL in each chamber on the day before exposure to GJ-EVs. GJ-EVs were dyed with SYTO^R^ RNASelect^TM^ green fluorescent cell stain (Invitrogen, Carlsbad, CA, USA) for 3 h and then observed with a laser microscope (Leica TCS SP8, Leica MYCROSYSTEMS, Wetzlar, Germany). PBS was used instead of EVs for control samples.

### 4.13. Terminology

MiRNA terminology used was according to proposed miRNA nomenclature guidelines [38], and gene symbols, according to the Human Genome Organization (HUGO) Gene Nomenclature Committee (HGNC) https://www.genenames.org/.

## Figures and Tables

**Figure 1 ijms-20-00953-f001:**
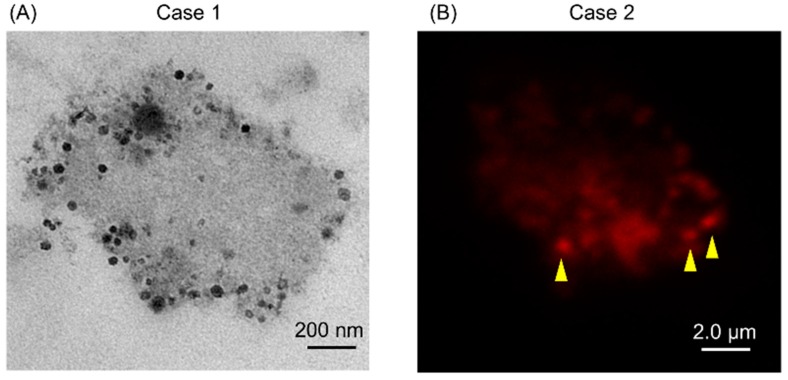
(**A**) Transmission electron microscopy (TEM) showed that extracellular vesicles (EVs) in gastric juice (GJ) of gastric cancer (GC) patient were covered with mucus. Two cases of GJ were examined; (**B**) CD63 was partially positive (yellow arrowhead) in GJ of GC. One case of GJ was examined.

**Figure 2 ijms-20-00953-f002:**
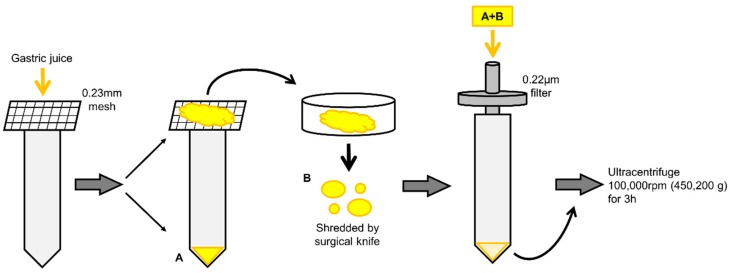
Illustration of preprocessing step to isolate EVs from GJ. First, GJ was filtered through a 0.23 mm-mesh for removing residues (**A**). The mucus was shredded by a surgical knife (**B**). The shredded mucus and filtrate were passed through a 0.22-μm pore filter (**A** + **B**).

**Figure 3 ijms-20-00953-f003:**
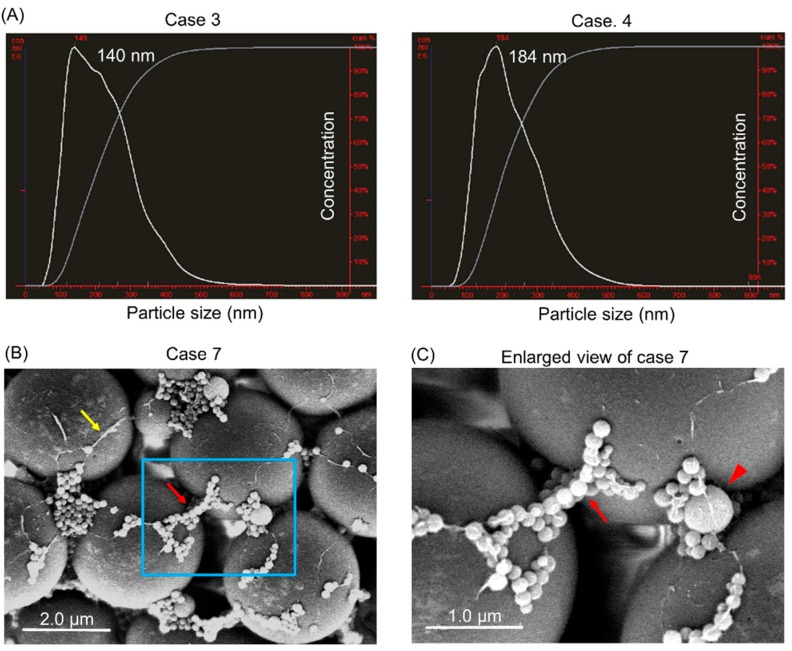
Isolates from GJ of GC patients contained EVs. (**A**) Size characterization of extracts from GJ by nanoparticle tracking analysis (NTA). The peak size of each sample is indicated in each graph. Four cases (cases 3–6) were examined, and representative graphs of cases 3 and 4 are shown; (**B**,**C**) Morphological study using scanning electron microscope (SEM). Three cases were examined. (**B**) SEM of the isolates from case seven. Red arrow indicate EVs on microbeads; and yellow arrow, mucus; area boxed in blue is enlarged in ‘’C’’. Scale bar = 2.0 µm; (**C**) Enlarged view of extracts is boxed in blue in (**B**). Red arrowhead: EVs more than 100 nm. Red Arrow: EVs less than 100 nm. Scale bar = 1.0 µm.

**Figure 4 ijms-20-00953-f004:**
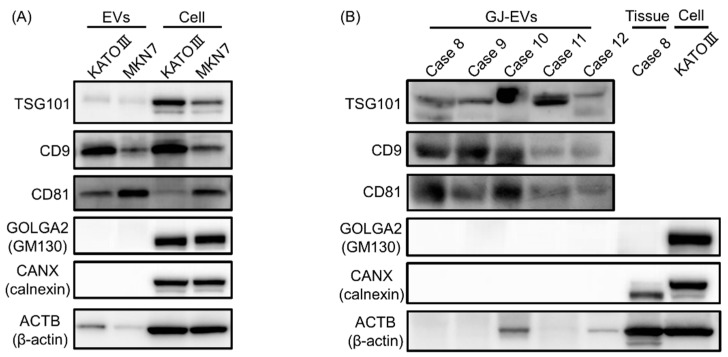
Protein markers of EVs were expressed in isolates from GJ. (**A**) Protein expression profiles of EVs from KATOIII and MKN-7 cell lines are shown. GOLGA2 (GM130) and CANX (calnexin) are used as negative indicator of EVs; (**B**) Those of the EVs of GJ from GC patients. Five cases (cases 8–12) of EVs from GJ were examined.

**Figure 5 ijms-20-00953-f005:**
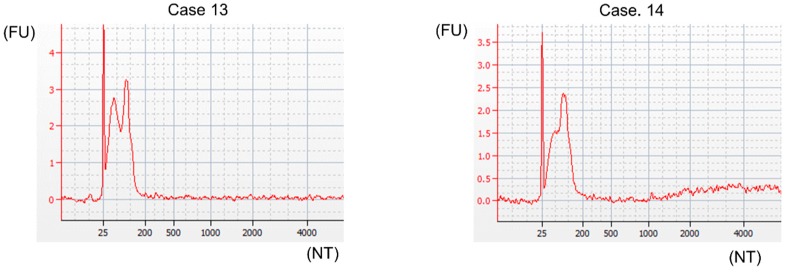
EVs from GJ contained miRNAs as genes. Characterization of RNA in EVs from GJ by Bioanalyzer. Four cases (cases 13–16) of EVs from GJ were examined. Representative graphs of two cases (cases 13 and 14) are shown.

**Figure 6 ijms-20-00953-f006:**
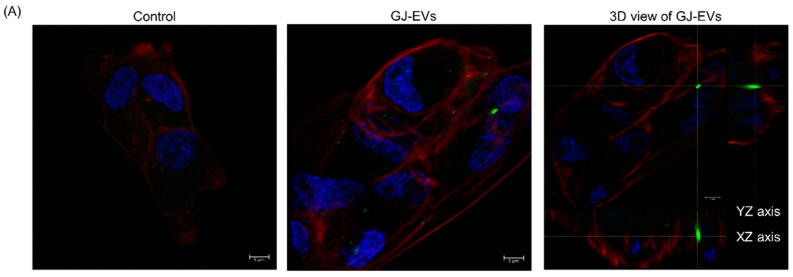
GJ-EVs from GC patients promoted the growth of fibroblast cells. Human normal fibroblastic ASF-4 cells were exposed to EVs extracted from KATOIII or the GJ of GC patients. (**A**) GJ-EVs (case 7′ sample) were dyed with a cell-permeating nucleic acid selective for RNA. The photographs were taken at 24 h after treatment. **Left** panel, control (phosphate-buffered saline (PBS)); **Middle** panel, GJ-EVs; **Right** panel, 3D view of the middle panel. RNA of GJ-EVs is dyed green; cell membrane, red; and nuclei, blue. Two cases of GJ-EVs were examined; (**B**) The cell growth of EV-treated ASF-4 cells. PBS was used as a control substance. The density of EVs was adjusted to 5000 (1.0) and 1250 (0.25) µg/mL. Two cases of GJ-EVs and EVs from KATOIII cells were examined. The effects were examined at 72 h after treatment. Results are presented as the mean ± standard deviation (SD); * *p* < 0.05; ** *p* < 0.01; *** *p* < 0.001; N.S., not significant; scale bar = 5 μm.

**Figure 7 ijms-20-00953-f007:**
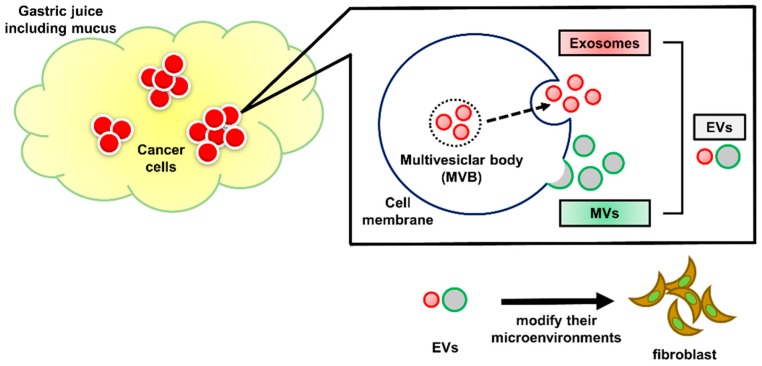
Schematic diagram of our study. GJ of GC patient contains EVs. GC cells partially construct their microenvironment such as fibroblasts through genes of GJ-EVs.

**Table 1 ijms-20-00953-t001:** Size distribution of extracellular vesicles (EVs) from gastric juice (GJ) of gastric cancer (GC) patients.

Case No.	Mean (nm)	Mode (nm)
3	225 ± 93	140
4	228 ± 86	184
5	270 ± 106	244
6	291 ± 138	212

**Table 2 ijms-20-00953-t002:** Data from the Agilent 2100 Bioanalyzer using the RNA 6000 Pico Kit.

Case	RNA Area	RNA Concentration	RNA Ratio	RNA Integrity Number (RIN)
13	39.2	293.2	0	2.6
14	33.2	248.3	0	2.6
15	1276.9	9552.4	0	2.6
16	16	119.7	0	2.7

**Table 3 ijms-20-00953-t003:** Threshold cycle (*C*_t_) value of each miRNA and snoRNA in EVs extracted from the GJ of GC patients.

Gene	Case 13	Case 14	Case 15	Case 16
*RNU6-6P*	34.47 ± 0.21	35.60 ± 0.26	35.82 ± 0.34	34.54 ± 0.17
MIRLET7A1-5p	28.11 ± 0.06	29.23 ± 0.12	26.07 ± 0.28	29.11 ± 0.16
MIR16-5p	24.79 ± 0.01	25.46 ± 0.04	14.14 ± 0.17	26.27 ± 0.11
MIR103a-3p	34.50 ± 0.18	32.80 ± 0.24	24.73 ± 0.15	31.90 ± 0.14
MIR191-5p	27.07 ± 0.02	27.22 ± 0.06	21.23 ± 0.27	29.46 ± 0.22
MIR423-5p	30.84 ± 0.47	34.46 ± 0.29	27.82 ± 0.08	34.56 ± 0.45

**Table 4 ijms-20-00953-t004:** Clinical and pathological features of gastric cancer patients.

Case	Age	Sex ^a^	Type ^b^	Size ^c^	Pathology ^d^	T ^e^	N ^f^	M ^g^	Stage ^h^
1	68	M	1	45 × 28	tub1, tub2 > pap	T2	N1	M0	pStageIIA
2	78	M	3	47 × 42	por2 > tub2	T3	N3a	M0	pStageIIIB
3	78	M	3	40 × 50	tub1, tub2, por	T3	N2	M0	pStageIIIA
4	66	M	0-IIc	5 × 5	tub2, por > sig	T1a	N0	M0	pStageIA
5	72	F	3	40 × 30	por2, por1	T3	N1	M0	ypStageIIB
6	36	F	4	49 × 38	por2, sig > tub2	T1b	N1	M0	ypStageIB
7	84	M	4	34 × 33	por2 >> tub2	T3	N1	M0	ypStageIIB
8	71	M	3	52 × 42	por1 > sig	T3	N0	M0	ypStageIIA
9	56	F	3	20 × 20	por2 > sig	T3	N0	M0	ypStageIIA
10	75	M	0-IIc	66 × 46	tub2, por2 > tub1	T3	N3a	M0	pStageIIIB
11	43	F	3	35 × 34	sig	T3	N1	M0	ypStageIIB
12	37	F	4	-	por, sig	T3	N0	M1	sStageIV
13	77	F	3	122 × 92	tub1, pap, tub2, por2	T4a	N3b	M0	pStageIIIC
14	54	F	4	49 × 46	por2 > sig	T4a	N0	M0	pStageIIB
15	50	M	4	-	por, sig	T4a	N2	M1	sStageIV
16	76	M	0-IIa + IIc	1.2 × 1.2	por2 > tub2 > tub1	T1b	N0	M0	pStageIA
17	78	M	0-I + Iia	53 × 35	tub1, pap > tub2	T1a	N0	M0	pStageIA
18	77	F	3	-	tub2, por	T4a	N1	M1	pStageIV

^a^ M, male; F, female; ^b^ Macroscopic classification; Type 1, mass type; Type 2, localized ulcerative type; Type 3, infiltrative ulcerative type; Type 4, diffuse infiltrating type; Type 5, unclassifiable; ^c^ Maximum diameter in mm. -, not measured; ^d^ Pathological classification; pap, papillary adenocarcinoma; tub 1, well-differentiated tubular adenocarcinoma; tub 2, moderately differentiated; por 1, poorly differentiated adsnocarinoma (solid type); por 2, (non-solid type); sig, signet-ring cell carcinoma; muc, mucinous adenocarcinoma; ^e^ Depth of tumor invasion; T1a, mucosa; T1b, submucosa; T2, Mucosa propria; T3, Subserosa; T4a, Serosa exposure; T4b, Serosa invasion; ^f^ lymph node metastasis; N0, no metastasis; N1, 1–2 metastases; N2, 3–6 metastases; N3a, 7–15 metastases; N3b, more than 16 metastases; ^g^ Distant metastasis; M0, no metastasis; M1 metastasis; ^h^ Progress degree; p, pathological stage; s, surgical stage; yp, stage after preoperative therapy.

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
