# Peer review of "Analysis of Extracellular Vesicles in Gastric Juice from Gastric Cancer Patients"

_ijms, 2019, doi:10.3390/ijms20040953_

Round 1

Reviewer 1 Report

Authors have addressed satisfactorily all previous comments made by the Reviewer.

Author Response

Thank you for your immediate review.

Reviewer 2 Report

Due to use of clinical materials, many limitations and issues still exist in this project. It should be addressed in a future study.

Most comments were addressed.

Quality of images in supplementary Fig S3 and Fig R1-6 is too poor. Please replace it to a good quality. 

Please check a spelling of  4-chanber slide in 310 line.

How many independent experiments are analyzed? This information needs to be in the figure and supple legends not just show representative numbers.

Author Response

Due to use of clinical materials, many limitations and issues still exist in this project. It should be addressed in a future study.

I agree with your thoughts. We will carry out continuing research, step by step.

Reviewer’s comment 1

Quality of images in supplementary Fig S3 and Fig R1-6 is too poor. Please replace it to a good quality. 

Our response to comment 1

As you pointed out, we rearranged the figures. Thank you for your suggestion.

Reviewer’s comment 2

Please check a spelling of 4-chanber slide in 310 line.

Our response to comment 2

We apologize for our carelessness. We corrected the spelling.

Reviewer’s comment 3

How many independent experiments are analyzed? This information needs to be in the figure and supple legends not just show representative numbers.

Our response to comment 3

Thank you for your insightful comment. Firstly, in order to establish the methods of isolation of EVs from gastric juice, we examined a lot of patient’s samples. After the establishment of the methods, the same tendency were observed in the most of samples by multilateral analysis. Certainly, the number of samples used in revised experiments were not many, because clinical materials were finite resources. However, all the results in additional experiments supported our previous results.

We added the descriptions regarding the numbers of each experimental samples. Also, we added a mention regarding the limitations of this study in the Discussion Section (line 184-185).

All your comment is scientifically correct. Thank you for your review.

Reviewer 3 Report

The authors have addressed my concerns.  Please ensure for the final version that the edited figures make it into the main document.  Right now the revised figures are at the end of the manuscript and the old versions are in the main body of the manuscript. 

Author Response

The authors have addressed my concerns. Please ensure for the final version that the edited figures make it into the main document. Right now the revised figures are at the end of the manuscript and the old versions are in the main body of the manuscript.

Thank you for your immediate reply. We rearranged our manuscript including revised figures through the editing by a staff of IJMS.

This manuscript is a resubmission of an earlier submission. The following is a list of the peer review reports and author responses from that submission.

Round 1

Reviewer 1 Report

Dear Authors,

The article entitled " Analysis of nanoparticles in gastric juice from gastric cancer patients" from Shuji Kagota, Kohei Taniguchi,  Sang-Woon Lee, Yuko Ito, Yuki Kuranaga,  Hiroyuki Hashiguchi, Yosuke Inomata, Ryo Tanaka, Keitaro Tashiro, Masaru Kawai, Yukihiro Akao and Kazuhisa Uchiyama  is well designed and well written. The Reviewer will recommend it for publication after some minor comments will be addressed by Authors.

Minor comments:

1- In Figure 3 C, Authors claimed that red arrows are showing exosomes. It would be good to see a better image because particles showed by red arrows appears to be around 20 nm (based on the scale provided). This is much smaller to the classical size of exosomes reported by Authors in the Introduction (50-100nm).

2- Based on the previous comment, the Reviewer suggest to the Authors to be cautious when they write line 94-95 " These findings suggested that our extracted nanoparticles from GJ contained both exosomes and SMVs"

3- line 89-90: The Authors wrote " ... and CD40 and integrin 5alpha, as markers of SMVs [17]". The reference provided does not clearly show that these markers are used to characterize SMVs. The Authors must provide better references to claim this point.

4- This is a side comment that Authors do not need to address, but it will be nice to discuss the fact that Case 15 is showing much higher levels of MIR16 -103 and -191 compare to Case 13 and 14 and make the correlation with data presented in Table 4.

5- typos:

    - line 83: "Aarrow" 

    - line 135  : space before "analysis"

<end of the report>                  

Reviewer 2 Report

This manuscript describes the methodology of purification and characterization of Extracellular vesicles (EVs) in gastric juice of gastric cancer patients. The study provides the basis for follow-up future studies related to the pathologic condition of gastric cancer (GC). Kagota et at al reported the ultracentrifuge isolation method combined with their own preprocessing step to isolate gastric juice-derived extracellular vesicles (GJ-EVs).  GJ-EVs were validated by NTA, SEM, western blotting and miRNA analysis.   However, there are multiple technological flaws and missing proper controls throughout the manuscript. These flaws are significant concerns to support their hypothesis that their EV isolation technique is dependable, and that GJ-EVs can be characterized as potential gastric cancer bio-markers.

The main concerns are listed below:

1.  The rationale that authors used gastric cancer patients’ sample is unclear. If authors intend to develop gastric juice-EVs derived from gastric cancer patients as gastric cancer specific marker, healthy control should be included.

2. It has been reported that the viscosity of clinical samples affects the efficiency of EV isolation. EV sedimentation efficiency decreases with an increase in viscosity of the sample (Please see ref: Methodological Guidelines to Study Extracellular Vesicles. Circ Res. 2017 May12; 120(10):1632-1648). Thus, viscous samples require higher-speed ultracentrifugation and longer time in comparison to centrifugation speed and time for EV isolation from cell culture supernatants. However, author used the same speed and time of centrifugation for EV isolation from cell culture supernatants and clinical samples in materials and method.

3. There are many issues reported regarding ultracentrifugation (UC) based EV isolation such as distortion of EV membrane and shapes due to high temperature and external forces. In addition, UC based EV isolation methods are highly susceptible to contaminants such as non-EV protein aggregates. UC is not compatible for routine clinical works. Other methods of EV isolation should be considered to confirm that the finding is not UC based artifacts.

4. Authors should clarify the term EVs vs. exosomes vs. SMVs. If isolated EVs fall into the category of both exosomes and SMVs, authors should only use term EVs” to be safe.

5. Line 244, exosomal RNA is not an appropriate term in this manuscript. If author use this term, they should isolate only pure exosomes without SMVs. There is no evidence that these RNAs were only being isolated from exosomes. Again, authors should  be used the term “EVs”.

6. Line 75:  authors reported that there are both exosomes and shedding microvesicles with the diameter size of 30-210 nm range (SMVs) in gastric juice (GJ). However, it is difficult to conclude that homogenous group of EVs (i.e. pure exosome population vs. pure SMVs). Exosomes should be distinguished from SMVs unless authors develop the immuno-affinity based isolation (i.e. CD63 Ab based pull-down of exosomes) to separate these two groups of EVs within the range of 30-210 nm diameter size.

7. Although there is no a gold-standard isolation method, analyzing proteins present in EVs need to follow minimal requirements of the International Society for Extracellular Vesicles (ISEV: J Extracell Vesicles. 2014 Dec 22;3:26913). Thus, non-EV markers such as GM 130 (Golgi marker) and Calnexin (ER marker) should be included to Figure 4 western blotting data to confirm that final EV preps are not contaminated with non-EV proteins. Whole cell lysates should also be used with exosomal markers (TSG, CD9, CD81and CD40) used in this figure.

8. In Figure 4 legends, β-actin cannot be used for negative control of nanoparticles. Again, author should confirm nanoparticles with using negative control markers such as GM 130. 

9. Line 110, six miRNAs cited may not represent universal EV miRNA markers based on heterogeneity of EVs from different organs. Comprehensive and comparative studies of miRNAs isolated from gastric juice and multiple other organs should be performed to identify suitable reference EV miRNAs.

10. In Figure 6, PBS could not be used as a negative control. GJ-EVs derived from normal healthy samples should be used as control.

11. In Figure 6, author need to show the evidence of GJ-EVs being incorporated into fibroblast cells. GJ-EVs should be labeled by dyes such as DIL to reach this conclusion.

12. In Figure 6, same numbers of EVs must be used into recipient cells (fibroblast cells). EVs concentration should be calculated by Nanosight. There is no mentioning of exact EV concentration of isolated samples throughout the manuscript.   % density is unacceptable quantification of EV concentration.

Minor comment:

1. Page 1, line 44: ref 12 is not a review.  Reference must be corrected.

2.  MIR16, MIR103, MIR191, MIR423 should be written as miR-16, etc.

Reviewer 3 Report

In this manuscript Kagota et al., purify extracellular vesicles (EVs) from the gastric juice of gastric cancer patients.  The authors use a variety of techniques including nanoparticle tracking analysis, immunoblotting of known EV markers and electron microscopy to validate that they are isolating EVs from this body fluid.  In addition, microRNAs that have previously been shown to be present in EVs from other biofluids were validated from the EVs from gastric juice. The work is timely and identifies EVs from a new body fluid, which may be relevant in future studies focused on using EVs as cancer biomarkers.  Overall, the manuscript is appropriate for this journal and this special issue.  There are some English grammar issues and the manuscript would benefit from additional editing. I have indicated some issues below in my comments. 

            In general, the nomenclature of both EVs and miRNAs could use changing.  The word “nanoparticles” should be changed to EVs, especially in the title.  miRNA nomenclature also needs to be clarified, especially with relation to which exact miRNA the authors are validating (for example miR-103 is not enough detail, there is miR-103a and miR-103b and also two mature sequences that are processed from each of these precursors). 

Here are also some issues that should be addressed:

Abstract line 18: nucleic acid should be plural “nucleic acids”

Line 24: change “extracted by the ultracentrifuge method to”…. “isolated by ultracentrifugation”

Line 42: The term EVs is a broad term and also encompasses apoptotic bodies.  This should also be listed along with microvesicles and exosomes. 

Don’t abbreviate shedding microvesicles as SMV  (this may be confused with small microvesicles).  Just abbreviate if needed as MV. 

Line 51: This sentence is awkward and needs rewording.  Justification in this paragraph needs some work.

Results:  The abstract and introduction were focused on EVs and then the authors switch to “nanoparticles”.  Keep with the term EVs.

Line 58: Instead of… “using the normal ultracentrifuge method” change to “ using ultracentrifugation” or “using the ultracentrifugation method”

Line 75:  You can’t conclude that they are exosomes or MVs.  You can just say based on size that they can be classified as exosomes and MVs.

Line 83: Aarrow should be Arrow

Pg. 98  B-actin is not a negative indicator of EVs.  It is a soluble protein that can be also be detected in EVs.  Please remove this sentence.

Is TSG TSG101?  If so, please indicate.

Line 102 Change …. “which are representative gene products of EVs” to “which are found in EVs” or “which are present in EVs”

Line 105.  Should be… “the Bioanalyzer”

Line 108: MIRLET7A1 is not the appropriate nomenclature for this miRNA.  Please confirm using the miRbase website the proper nomenclature. 

miRNAs are not italicized and should be written as miR-16 etc…  In addition, many of these miRNAs have multiple miRNAs that named similarly.  For example miR-103 has a miR-103a and miR-103b and also further has nomenclature to indicate what strand it is coming from ie. miR-103a-1-5p.  It is essential to indicate properly which miRNA you are examining for consistency and reproducibility. 

Line 110:  change “test genes” to “test miRNAs”

Line 111  miRNAs are not strongly expressed in EVs.  They would have higher levels or present in EVs.

Line 112 and elsewhere.  Gene products is not an appropriate term.  Either use genes or miRNAs.   

Line 182:  Not sure that Clinicopathological is the appropriate term.

Line 195:  Ultracentrifugation is typically at 100,000 x g.  It is indicated in rpm, is this correct?  Please convert and note that if it is not in g, this is NOT considered ultracentrifugation. 

Line 206:  “Extracts” do you mean EVs extracted? 

Line 219: The version of the NanoSight should be indicated.  Please also indicate the camera level, detection level and number of videos and length were used for the measurements.